# OpenReview forum: "Emotional Manipulation is All You Need: A Framework for Evaluating Healthcare Misinformation in LLMs"
_ICLR.cc/2025/Workshop/BuildingTrust — Submitted to BuildingTrust_

### Official Review · Reviewer_faXi · 2025-03-01

**Rating:** 4
**Confidence:** 3

**Review:**

# Summary
The paper investigates vulnerabilities in healthcare LLMs by examining prompt injection attacks—particularly those that leverage emotional manipulation—to assess their potential to amplify harmful medical misinformation. It evaluates 8 state-of-the-art LLMs across 112 attack scenarios.

# Strengths
The paper combines 6 prompt injection techniques with both emotional and non-emotional variants, offering a clear view of model performance. It provides categorization for both response severity and cancer treatment misinformation.

# Questions & Weakness:
1. Could you please clarify whether the response severity and cancer treatment misinformation categorizations are performed by LLMs acting as judges or by human expert judges? This detail is not clearly mentioned in the Methods section.

2. With 112 test scenarios (derived from 6 attacks + 1 baseline, times 8 LLMs and 2 variants), is only one prompt used per test scenario? This limited scale might impact the reliability of the conclusions.

3. It appears that both the attack techniques and the categorizations are based on previous work. Could you explain the source of the medical prompts and clarify any novel contributions of this paper?

4. The abstract mentions crucial insights for developing robust safety mechanisms before clinical deployment. However, these insights are not clearly detailed in the manuscript.

5. If the goal is to identify AI safety risks in healthcare, a more effective approach might be to categorize risky inputs within healthcare scenarios (attacking methods and emotional manipulation are more like perturbations). The current benchmark's usage in this context is unclear.

---

### Official Review · Reviewer_yS4z · 2025-03-02
**The paper lacks clarity on its methodology, threat model, dataset, and evaluation of helpfulness, and does not align with its title or focus on healthcare-specific LLMs**

**Rating:** 3
**Confidence:** 3

**Review:**

In the paper, the authors claim to conduct a systematic evaluation of healthcare misinformation in LLMs. This is an interesting research avenue and requires collaboration within the research community to develop more robust LLMs for the healthcare domain.

However, I believe the paper has major flaws. The paper's title does not accurately reflect the content of the paper. Additionally, I found the attack threat model to be unhelpful—why would anyone use jailbreak prompts when seeking medical advice from LLMs? The threat model could have been discussed more thoroughly.

Another major issue is the lack of clarity regarding the dataset used for the results. It is unclear whether the results were based solely on the examples provided in the Appendix with different prompts, or if other datasets or prompts were used, which is not specified.

If the paper aims to focus on the safety of healthcare LLMs, it could have been more impactful if it had tested LLMs specifically trained for healthcare tasks, such as Med-based LLMs.

Also, the paper fails to mention how it measures whether the generated responses are helpful or harmful. The categorization in Table 1 alone is not sufficient to evaluate the helpfulness or harmfulness of the medical advice.

---

### Decision · Program_Chairs · 2025-03-04

Reject